# Afferent convergence to a shared population of interneuron AMPA receptors

Reagan L. Pennock [1], Luke T. Coddington [1,2], Xiaohui Yan [1], Linda Overstreet-Wadiche [1] ✉ & Jacques I. Wadiche [1] ✉

Precise alignment of pre- and postsynaptic elements optimizes the activation of glutamate receptors at excitatory synapses. Nonetheless, glutamate that diffuses out of the synaptic cleft can have actions at distant receptors, a mode of transmission called spillover. To uncover the extrasynaptic actions of glutamate, we localized AMPA receptors (AMPARs) mediating spillover transmission between climbing fibers and molecular layer interneurons in the cerebellar cortex. We found that climbing fiber spillover generates calcium transients mediated by $Ca^{2+}$-permeable AMPARs at parallel fiber synapses. Spillover occludes parallel fiber synaptic currents, indicating that separate, independently regulated afferent pathways converge onto a common pool of AMPARs. Together these findings demonstrate a circuit motif wherein glutamate 'spill-in' from an unconnected afferent pathway co-opts synaptic receptors, allowing activation of postsynaptic AMPARs even when canonical glutamate release is suppressed.

Alignment of pre- and postsynaptic elements of glutamatergic synapses ensures that glutamate released from an active zone efficiently activates apposed postsynaptic receptors[1–3]. The specificity of excitatory synaptic connectivity is reinforced by the rapid rise and fall of glutamate concentrations within the synaptic cleft following vesicular release[4,5]. It is generally accepted that low concentrations of glutamate outside the synapse generate only weak receptor activation, supporting the idea of synapse independence[6]. Such strict synaptic specificity underlies current efforts to define the circuits of information transfer by mapping anatomically defined synaptic connectivity[7–10]. However, across brain regions, there are many examples of glutamate actions outside of the synaptic cleft, and the physiological and pathological role of such glutamate "spillover" remains debated[11].

A large literature supports synapse specificity mediated by low-affinity AMPARs, whereas higher-affinity NMDARs can respond to low levels of glutamate at a distance from the presynaptic release site[12–17]. Even so, intense synaptic activity, densely-spaced release sites, or multivesicular release can also effectively activate AMPA receptors (AMPARs) distant to presynaptic release sites[18–24]. Since a fraction of AMPARs resides in the extrasynaptic membrane awaiting integration into postsynaptic densities[25,26], AMPAR-mediated spillover responses may be analogous to volume transmission from inhibitory neurogliaform cells that recruit extrasynaptic GABARs[27–29]. Alternatively, glutamate spillover may activate neighboring synaptic AMPARs, an important caveat for synapse independence, as shown at specialized glomerular synapses[19,30]. Visualization of glutamate spread using optical sensors and modeling studies have questioned the assumptions of synapse independence[11,31]. Understanding the spatial organization of AMPARs that respond to the extrasynaptic spread of glutamate is important to address the consequences for synapse specificity.

Molecular layer interneurons (MLIs) provide a model system to address the location of AMPARs mediating glutamate spillover. MLIs receive 'simple' canonical synapses from parallel fibers (PFs)[32] and also respond to glutamate spillover from climbing fibers (CFs) in the absence of anatomically defined synaptic junctions[18,20,21,33]. PF and CF EPSCs can be independently isolated by several criteria in vitro[18,20,21,34] and CF spillover has been identified in vivo, potentially playing a role in MLI plasticity[35–37]. Here we use two-photon (2 P) $Ca^{2+}$ imaging to localize AMPARs that mediate CF spillover to MLIs. We show that glutamate from CFs activates $Ca^{2+}$-permeable (CP) AMPARs at PF synapses, demonstrating that spillover and synaptic responses from distinct

[1]Department of Neurobiology, University of Alabama at Birmingham, Birmingham, AL 35294, USA. [2]Present address: Howard Hughes Medical Institute Janelia Research Campus, Ashburn, VA 20147, USA. ✉e-mail: lwadiche@uab.edu; jwadiche@uab.edu

afferent pathways are mediated by the same population of post-synaptic receptors. Interestingly, PF but not CF activation of CP-AMPARs is suppressed by presynaptic GABA$_B$R activation, suggesting that GABAergic activity selectively modulates PF but not spillover transmission. This afferent convergence represents a strategy for maintaining AMPAR activation in the absence of synaptic release and without the formation and maintenance of conventional synaptic connections.

## Results

### Mapping CF-evoked Ca²⁺ transients on MLI dendrites

We recorded from MLIs in parasagittal slices containing the cerebellar vermis and evoked glutamate spillover from nearby CF-PC synapses. We isolated AMPAR-mediated responses in the presence of the NMDAR antagonist (R)-CPP (5 µM) and the GABA$_A$R antagonist picro-toxin (100 µM) using an intracellular solution that contained Alexa 594 (30 µM) and Fluo-5F (200 µM) to visualize cell morphology and localize sites of Ca²⁺ influx, respectively. We stimulated CFs with a theta glass

electrode placed near the Purkinje cell layer (PCL; Fig. 1a and Supplementary Fig. 1) and identified spillover EPSCs by strong paired-pulse depression (Supplementary Fig. 1b, e) and an all-or-none response to changes in stimulus intensity (Fig. 1c–e)[18,20,21]. After isolating a CF-MLI spillover response, we used two-photon imaging (2 P) to search for sites of Ca²⁺ influx by scanning at 2–5 Hz while stimulating CFs (0.1 or 0.05 Hz, Fig. 1). We detected transient increases in Fluo-5F fluorescence coincident with CF stimulation within restricted segments of the dendritic arbor. We characterized these CF-evoked Ca²⁺ transients (CaTs; Fig. 1a, b) using line scans as previously described for spatially restricted CaTs arising from CP-AMPAR synapses on aspiny interneurons[38,39]. To confirm that CaTs arise from CFs, we tested that they share the hallmark characteristics of CF-MLI EPSCs, i.e., paired-pulse depression and an all-or-none response to changes in stimulation intensity. Increasing the line scan sampling rate from 0.5 to 1 kHz allowed measurement of paired-pulse depression, which was comparable to that of the EPSC (PPR$_{CaT}$ = 0.18 ± 0.053 vs. PPR$_{EPSC}$ = 0.26 ± 0.015, $p$ = 0.20, paired t-test; Supplementary Fig. 1b,

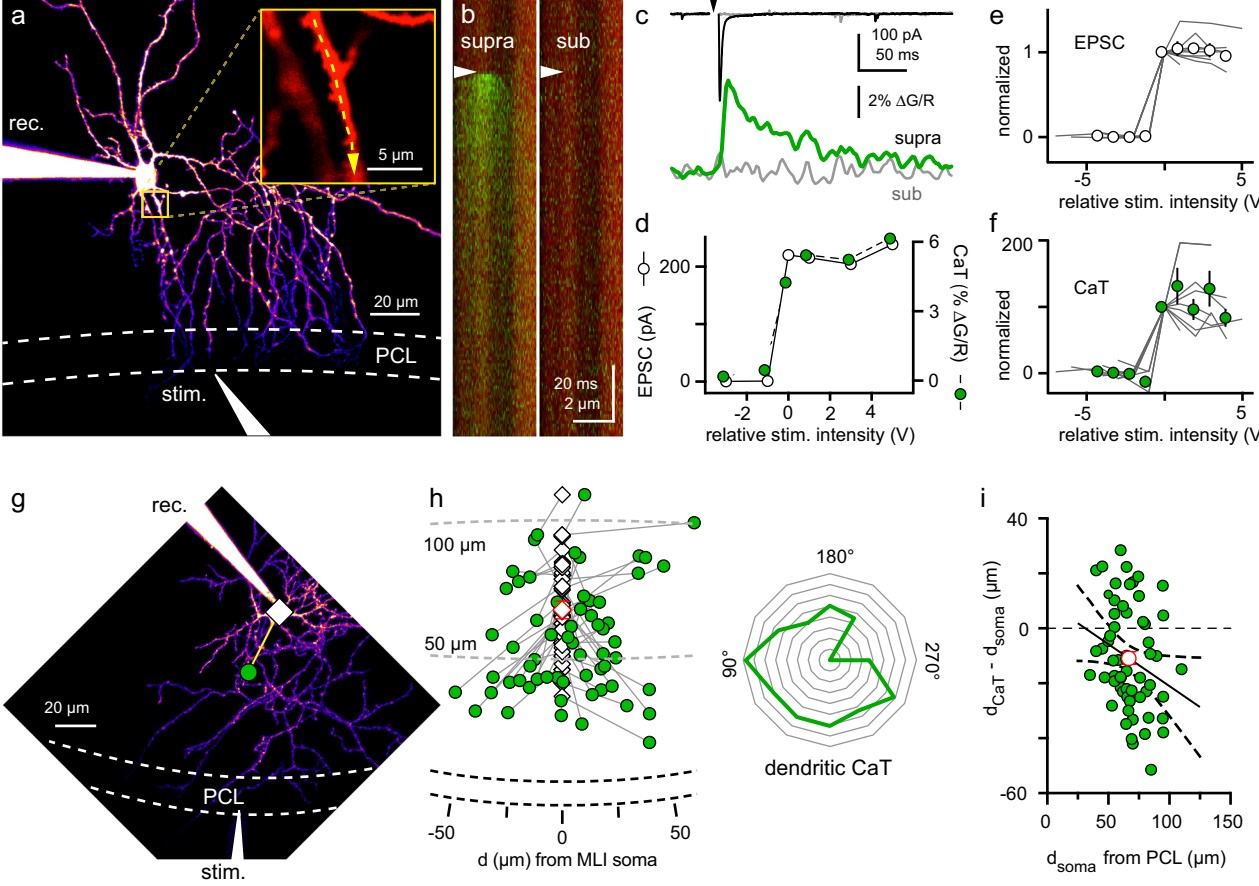

**Fig. 1 | Localization and mapping of CF-evoked Ca²⁺ transients. a** 2 P maximum Z-projection of an MLI filled with Alexa 594 (30 µM) via the recording (rec.) pipette. The stimulating electrode (stim.) was also visualized with Alexa 594 (5 µM). Inset: magnified boxed region showing dendritic line scans (dashed line) used to image CF-evoked CaTs. **b** Overlaid scans of red (Alexa 594) and green channel (Fluo-5F; 200 µM) from (A) during a supra- (left) and subthreshold (right) stimulation (arrowhead). **c** Average CF EPSC (top) and CaT (bottom, green) in response to supra- and subthreshold stimuli (gray traces). **d** Peak amplitudes of CF EPSCs (white circles) and CaTs (green circles) versus stimulus intensity from **a**–**c**. EPSCs and CaTs exhibit an all-or-none response with the same threshold. **e**, **f** Peak EPSCs and CaT amplitudes versus relative stimulus intensity (*n* = 9). 0 V represents the stimulus threshold for each experiment (gray lines). **g** Example location of a CF CaT (green circle) relative to the MLI soma (diamond). The distance and angle of the CaT relative to the soma was measured by drawing a line between the two points. **h** (left)

Summary of all CaTs (green) and respective MLI somas (diamonds, *n* = 62 cells). The average distance of MLI somas was 67 ± 2.0 µM from the PCL (red diamond). **h** (right) Radial histogram shows the relative frequency and projection angles of dendrites with a CF-evoked CaT. **i** The direction and distance of a CF-evoked CaT (relative to the MLI soma) versus the distance of the MLI soma from the PCL. Values >0 project away from the PCL, whereas values <0 project toward the PCL. Linear regression (line with 95% CIs dashed lines) shows that MLIs located further from the PCL are more likely to have CF-evoked CaTs located on dendrites projecting towards the PCL (slope = −0.3, $R^2$ = 0.06, $p$ = 0.04). On average, CF-evoked CaTs were located 11 ± 2.4 µm closer to the PCL than the soma (red circle). R-CPP (5 µM) and picrotoxin (100 µM) were included in all recordings. Data are shown as mean ± SEM. Source data are provided in the Source Data file: Source Data Figure1.xlsx.

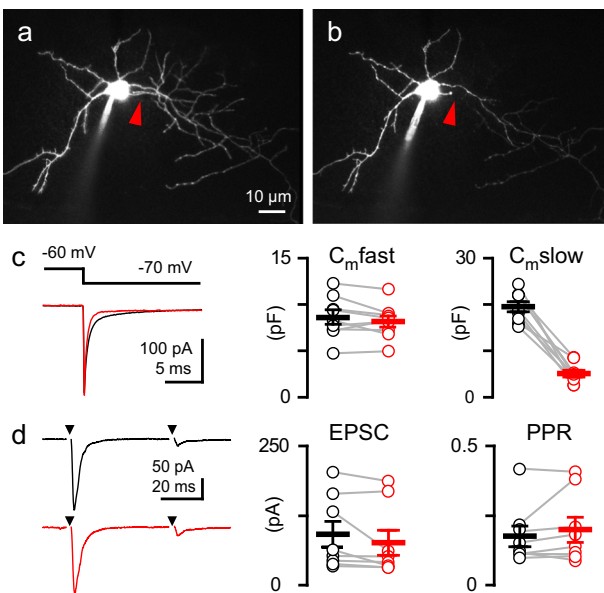

**Fig. 2 | MLI axons are not necessary for CF EPSCs.** Example of an Alexa 594 filled MLI before (**a**) and 20 min after (**b**) axotomy at the site indicated by an arrow. **c** (left) Average current in response to a −10 mV voltage step before (black) and after (red) axotomy. **c** (right) Axotomy did not alter the fast component of the membrane capacitance ($8.6 \pm 0.75$ to $8.2 \pm 0.61$ pF, $p = 0.2$, paired $t$-test) but reduced the slow component ($19.8 \pm 1.1$ to $5.4 \pm 0.73$ pF, $p < 0.001$, Two-tailed paired $t$-test, $n = 9$). **d** (left) Examples CF EPSCs in response to paired-pulse stimulation (50 ms) before (black) and after (red) axotomy. **d** (right) Axotomy did not alter CF EPSC amplitude ($92 \pm 23$ to $77 \pm 22$ pA, $p = 0.12$, Two-tailed paired $t$-test) or paired-pulse ratio ($0.18 \pm 0.03$ to $0.20 \pm 0.04$ PPR, $p = 0.69$, Two-tailed paired $t$-test, $n = 9$). Data are shown as mean ± SEM. Source data are provided in the Source Data file: Source Data Figure2.xlsx.

e). In contrast, CaTs resulting from activation of PF synapses showed paired-pulse facilitation like that of PF EPSCs ($\text{PPR}_{CaT} = 1.1 \pm 0.18$ vs. $\text{PPR}_{EPSC} = 1.5 \pm 0.045$, $p = 0.06$, paired $t$-test; Supplementary Fig. 1c–e). CF-evoked CaTs were also all-or-none with the same threshold as the CF EPSC (Fig. 1b-f). The amplitude of PF EPSCs increased with stimulation intensity while individual PF-evoked CaTs were all-or-none (see below), as previously reported[40]. The AMPAR antagonist NBQX (10 μM) completely blocked EPSCs and CaTs evoked by either CFs or PFs, confirming that both pathways are mediated by AMPARs (Supplementary Fig. 1). Glutamate released from CFs onto Purkinje cells can extend to mGluRs[41,42] and induce mGluR1-mediated CaTs when glutamate reuptake is blocked[43], and can generate mGluR-mediated CaTs that are blocked by NBQX[44]. MLIs also express CPCCOEt-sensitive mGluR1 currents[45], so we tested whether CPCCOEt (100 μM) affected CF-evoked CaTs. However, CPCCOEt had no effect (Supplementary Fig. 1g), ruling out a contribution of mGluR1. Together, these findings demonstrate CaTs evoked by CF or PF stimulation reflect the properties of the underlying EPSCs and can be reliably isolated from one another.

PFs run orthogonal to the plane of parasagittal slices[46], making it easy to localize active PF-MLI synapses along dye-filled MLI dendrites directly below the stimulating electrode (Supplementary Fig. 1c). In contrast, the location of CF-evoked CaTs was harder to predict, since CFs branch to form hundreds of synapses along the dendrites of PCs[46]. To determine whether there was a pattern to the location of spillover sites on MLI dendrites, we acquired Z-series of dye-filled MLIs to map the location of CF-evoked CaTs relative to the soma and PCL ($n = 62$; Fig. 1g). Since MLIs primarily project processes in the longitudinal plane with little projection in the transverse plane[47], we localized spillover sites in two dimensions. CaTs were detected throughout MLI

dendritic arbors within the inner two-thirds of the ML, consistent with the extent of CF innervation of PCs (Fig. 1h). The somas of MLIs on which CF-evoked CaTs were located were, on average, $67 \pm 2.0$ μm from the PCL (Fig. 1h). On average, CaTs were located on a segment of dendrite projecting slightly towards the PCL ($56 \pm 3.0$ μm from PCL; Fig. 1i). The further an MLI soma was from the PCL, the more likely the CF-evoked CaT was found on a section of dendrite projecting towards the PCL, consistent with CF innervation of PCs (slope = −0.3, $R^2 = 0.06$, $p = 0.04$, linear regression; Fig. 1i). There was no correlation between location and amplitude of CaTs ($p = 0.06$, $R^2 = 0.06$, linear regression; Supplementary Fig. 2a) or the corresponding EPSC ($p = 0.84$, $R^2 < 0.001$, linear regression; Supplementary Fig. 2b), implying that the amount of spillover received by MLIs is independent of the dendritic location of the spillover site within the ML.

### Axonal AMPARs do not contribute to CF EPSCs
In addition to dendritic AMPARs, there is evidence that MLI axons express GluA2-containing receptors that are activated during CF activity to regulate GABA release[48–50]. To test the possibility that focusing on MLI dendrites excludes a contribution of axonal AMPARs to CF spillover responses, we assayed CF EPSCs before and after two-photon illumination-based axotomy[51]. After establishing a baseline CF spillover EPSC, we cut visually identified axons <30 μm from the soma with 5–10 high-intensity line scans (Fig. 2a, b). We verified successful axotomy by a reduction of the slow axonal capacitance component of the voltage step response[51], which decreased from $19.8 \pm 1.1$ pF to $5.4 \pm 0.7$ pF ($n = 9$, Fig. 2c). Importantly, MLI axotomy did not significantly change either the CF EPSC amplitude or paired-pulse ratio (amplitude: $92.1 \pm 20.4$ to $78.3 \pm 19.8$ pA, $p = 0.11$; PPR: $0.18 \pm 0.03$ to $0.19 \pm 0.04$, $p = 0.69$, $n = 9$ for each; Fig. 2d). This shows that AMPARs located on MLI axons do not contribute to the CF-mediated spillover EPSC and thus we focused exclusively on dendritic EPSCs and CaTs.

### Ca²⁺-permeable AMPARs mediate spillover CaTs
CF- and PF-evoked EPSCs and CaTs were completely blocked by the AMPAR antagonist NBQX, suggesting that CaTs evoked by both pathways are mediated by CP-AMPARs. Accordingly, the use-dependent CP-AMPAR blocker NASPM (50–100 μM) strongly inhibited CaTs evoked by CF stimulation ($0.10 \pm 0.011$ to $0.029 \pm 0.0048$ ΔG/R; $p < 0.0001$; paired $t$-test) and PF stimulation ($0.088 \pm 0.0089$ to $0.021 \pm 0.0067$ ΔG/R; $p < 0.0001$; paired $t$-test), and inhibited CaTs from both pathways to a similar extent ($\text{CaT}_{NASPM}/\text{CaT}_{baseline}$: CF vs. PF, $0.29 \pm 0.049$ vs. $0.22 \pm 0.051$; $p = 0.65$, Tukey's multiple comparison test; Fig. 3). NASPM also inhibited EPSCs evoked from CF spillover ($430 \pm 105$ to $175 \pm 36.0$ pA; $p = 0.01$; paired $t$-test) and PF synapses ($324 \pm 32.1$ pA to $67.3 \pm 10.8$ pA; $p = 0.0001$; paired $t$-test), with CF spillover EPSCs inhibited to a lesser extent ($\text{EPSC}_{NASPM}/\text{EPSC}_{baseline}$: CF vs. PF, $0.44 \pm 0.044$ vs. $0.21 \pm 0.025$; $p = 0.02$; Tukey's multiple comparison test; Fig. 3d). It is well established that PF EPSCs (and resulting CaTs) are mediated by synaptic CP-AMPARs[39,52]. Since glutamate spillover is typically associated with the activation of extrasynaptic receptors and extrasynaptic AMPARs in MLIs are Ca²⁺-impermeable[52], but see ref. 53, we were surprised by the robust CaTs and NASPM sensitivity of CF responses. Together these results suggest the possibility that CF spillover is mediated by AMPARs at PF synapses, providing an example of glutamate 'spill-in' wherein glutamate released from one afferent pathway targets receptors in a separate pathway. In this case, a common pool of AMPARs is activated by local (PF) or distant (CF) presynaptic release sites.

### Spillover CaTs exhibit synapse-like compartmentalization
To begin testing the idea that CF spillover activates synaptic CP-AMPARs, we compared the spatial extent of CF and PF-evoked CaTs. CF stimulation often evoked multiple CaTs on a stretch of a dendrite with the same threshold as the CF-evoked EPSC (Fig. 4a–c). To measure the

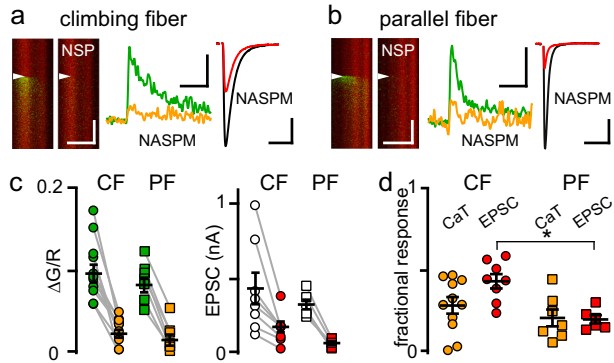

**Fig. 3 | NASPM inhibition of EPSCs and CaTs. a** (left) Average CF line scans and CaTs before (green) and after (orange) the $Ca^{2+}$-permeant AMPAR blocker NASPM (100 µM). **a** (right) Average CF EPSC before (black) and after (red) NASPM. **b** Similar images and traces for PF CaTs and EPSCs. Scale bars: image (100 ms, 3 µm), CaT (5% ΔG/R, 50 ms), EPSC (100 pA, 10 ms). **c** (left) NASPM block of CF- and PF-evoked CaTs (CF: $0.10 \pm 0.011$ to $0.029 \pm 0.0048$ ΔG/R, $n = 11$, $p < 0.001$, Two-tailed paired $t$-test; PF: $0.088 \pm 0.0089$ to $0.021 \pm 0.0067$ ΔG/R, $n = 7$, $p < 0.001$, Two-tailed paired $t$-test). **c** (right) NASPM block of CF- and PF EPSCs (CF: $431 \pm 105$ to $175 \pm 36.0$ pA, $n = 8$, $p = 0.01$, Two-tailed paired $t$-test; PF: $324 \pm 32.1$ to $67.3 \pm 10.8$ pA, $n = 6$, $p < 0.001$, Two-tailed paired $t$-test). **d** Comparison of NASPM block of CF and PF responses. NASPM blocked CF EPSCs less than PF EPSCs ($0.44 \pm 0.044$ vs. $0.21 \pm 0.025$, $n = 8$ and 6), $p = 0.02$, Tukey's multiple comparisons test. Asterisk denotes $p < 0.05$. Data are shown as mean ± SEM. Source data are provided in the Source Data file: Source Data Figure3.xlsx.

distance between neighboring CaTs, we fit the average fluorescence profile at the peak (3–9 ms post-stimulus) with a sum of Gaussian functions (Fig. 4d)[54]. While some CF-evoked CaTs were widely spaced and easily distinguished from one another (Fig. 4d left), the majority were close together (Fig. 4d right), reminiscent of the distance between PF-MLI synapses[39]. We thus compared the fluorescence profile of closely spaced CF-evoked CaTs with those recruited by activating bundles of PFs. Unlike CF-evoked CaTs, individual PF-evoked CaTs had varying stimulus thresholds and the amplitude of the EPSC was graded with stimulus intensity (Fig. 4e–g). Fluorescence profiles of neighboring PF-evoked CaTs showed similar spacing compared to CF-evoked CaTs (distance to nearest site, PF: $3.1 \pm 0.26$, CF: $3.2 \pm 0.35$ µm, $p = 0.41$, Mann–Whitney test; Fig. 4h, i).

PF-MLI synapses, like glutamatergic synapses on aspiny interneurons of the cortex, produce CaTs that are contained within submicron domains of dendrites[38,39]. This compartmentalization can be measured by fitting the fluorescence profile of line scans to Gaussians and determining the width (σ) as a function of time[38,39], assuming sufficient resolution of the 2P PSF (Supplementary Fig. 3). We compared the compartmentalization of individual CaTs near the temporal peak in fluorescence (5 ms post-stimulus) and found that both PF and CF-evoked CaTs had similar σ values (PF: $0.87 \pm 0.14$ µm, CF: $0.80 \pm 0.089$ µm; $p = 0.66$, unpaired $t$-test, Fig. 5a, b). The spatial spread of CaTs increased with the same time course (τ, $p = 0.97$, unpaired $t$-test) and reached a similar maximum value ($\sigma_{max}$; $p = 0.72$, unpaired $t$-test; Fig. 5c, d). This compartmentalization is slightly narrower than previously reported at room temperature[38,39]. These results show that the $Ca^{2+}$ influx evoked by CF spillover is contained within microdomains like those seen at PF synapses. The presence of multiple closely spaced CF-evoked CaTs shows that CF spillover can encompass more than one microdomain.

## CF spillover recruits AMPARs at PF synapses

A parsimonious explanation for the confinement of CF-evoked CaTs to synapse-like microdomains is that glutamate released from CFs 'spills in' to PF synapses. In this case, PF-evoked CaTs should also occur at CF

sites and PF EPSCs will be occluded when CF spillover occurs near PF synapses.

We first tested whether PF and CF stimulation can evoke CaTs at the same (overlapping) sites using alternating stimulation combined with 2P imaging. After isolating a CF EPSC and localizing a corresponding CaT (Fig. 6a, b), we activated nearby PFs using a second stimulating electrode (Fig. 6a, c). We integrated both CaT fluorescence profiles (peak at 3–9 ms post-stimulus) to calculate the cumulative ΔG/R distribution and compared them using a Kolmogorov–Smirnov test (Supplementary Fig. 4a, b). Sites were considered overlapping if the cumulative distributions were the same ($p > 0.05$; Supplementary Fig. 4c). We also used the normalized Gaussian fits of peak CaTs from both pathways to determine the distance between sites (Fig. 6b, c). This analysis revealed CF- and PF-evoked CaTs could occur at the same sites (distance between peaks: $0.31 \pm 0.05$ µm, Fig. 6d). At overlapping sites, the average amplitude of CF CaTs was slightly larger than PF CaTs (Supplementary Fig. 5a). This was surprising considering that peak [glutamate] resulting from CF spillover is reported to be lower than at PF synapses, at least in young rats[18]. To determine the relative [glutamate] resulting from CF- and PF-evoked release in mice, we applied the low-affinity AMPAR antagonist kynurenic acid (KYN; 500 µM) while stimulating either pathway. Inhibition of CF EPSCs by KYN was greater than PF EPSCs, indicating a higher [glutamate] underlying PF EPSCs (Supplementary Fig. 5b). We next determined the ratio of the CF and PF CaTs to the charge of their corresponding EPSCs at overlapping sites in experiments where PF EPSCs were recruited with near minimal stimulation. This analysis showed that the CaT/EPSC ratio is greater for the PF pathway (10/10 experiments), suggesting that CP-AMPARs mediate a larger fraction of PF EPSCs in comparison to CF EPSCs (Supplementary Fig. 5c).

In some experiments, a CaT near the overlapping site was recruited by activation of either pathway, i.e., Fig. 6b shows a second site activated by the CF near the overlapping site (Fig. 6e). The distance between the overlapping site and the non-overlapping site was greater than the distance between overlapping sites ($3.2 \pm 0.3$ µm vs. $0.31 \pm 0.05$ µm, $n = 24$, 23 sites; $p < 0.0001$, Dunnett's T3 multiple comparison test; Fig. 6e) and was consistent with the distance between neighboring CF or PF sites ($3.2 \pm 0.3$ µm vs. $3.2 \pm 0.4$ vs. $3.1 \pm 0.3$ µm, $p > 0.99$ for all comparisons, Dunnett's T3 multiple comparison test, from Figs. 4i, 6f). While these results show that CF spillover CaTs occur at PF synaptic sites, they do not address whether these two modes of transmission share a common pool of synaptic AMPARs.

To next test whether CF and PF afferents share AMPARs, we asked whether PF EPSCs are occluded by CF spillover at overlapping sites. If they share AMPARs, PF EPSCs following spillover will be reduced because AMPARs are occupied by glutamate released from CFs[55,56]. First, we used a two-pathway protocol to alternate stimulation of control PF ($PF_{alone}$) and CF ($CF_{alone}$) EPSCs at overlapping or non-overlapping sites, as described above. Then we stimulated PFs at varying interstimulus intervals (ISI: 1.3–100 ms) following the CF to generate a compound EPSC (CF + PF). Subtraction of the control CF EPSC from the compound EPSC allowed reconstruction of the subtracted PF component ($EPSC_{subt.}$; Fig. 7a, b). Recruitment of CF and PF EPSCs at overlapping sites (distance: $0.47 \pm 0.085$ µm; $n = 6$; Supplementary Fig. 6a, c) resulted in an $EPSC_{subt.}$ that was reduced by ~50% of control ($PF_{alone}$) at an ISI of 1.3 ms ($96 \pm 11$ pA to $54 \pm 8.3$ pA; $n = 10$; $p < 0.001$; paired $t$-test) and recovered with a time constant of 5.5 ms (Fig. 7a). In contrast, at non-overlapping PF and CF sites (distance: $4.1 \pm 0.95$ µm; $n = 6$; Supplementary Fig. 6b, c), there was little difference in the $EPSC_{subt.}$ compared to $PF_{alone}$ EPSC (Fig. 7b; $81 \pm 10$ pA to $71 \pm 10$ pA; $n = 6$; $p = 0.05$, paired $t$-test). Likewise, repeating the experiment without visualizing CaTs (blind) also resulted in little difference between the $EPSC_{subt.}$ compared to $PF_{alone}$ EPSC (Fig. 7c; $108 \pm 11.1$ pA to $102 \pm 15.2$ pA; $n = 7$; $p = 0.46$, paired $t$-test). Altogether, only the overlapping sites exhibited significant PF EPSC occlusion after

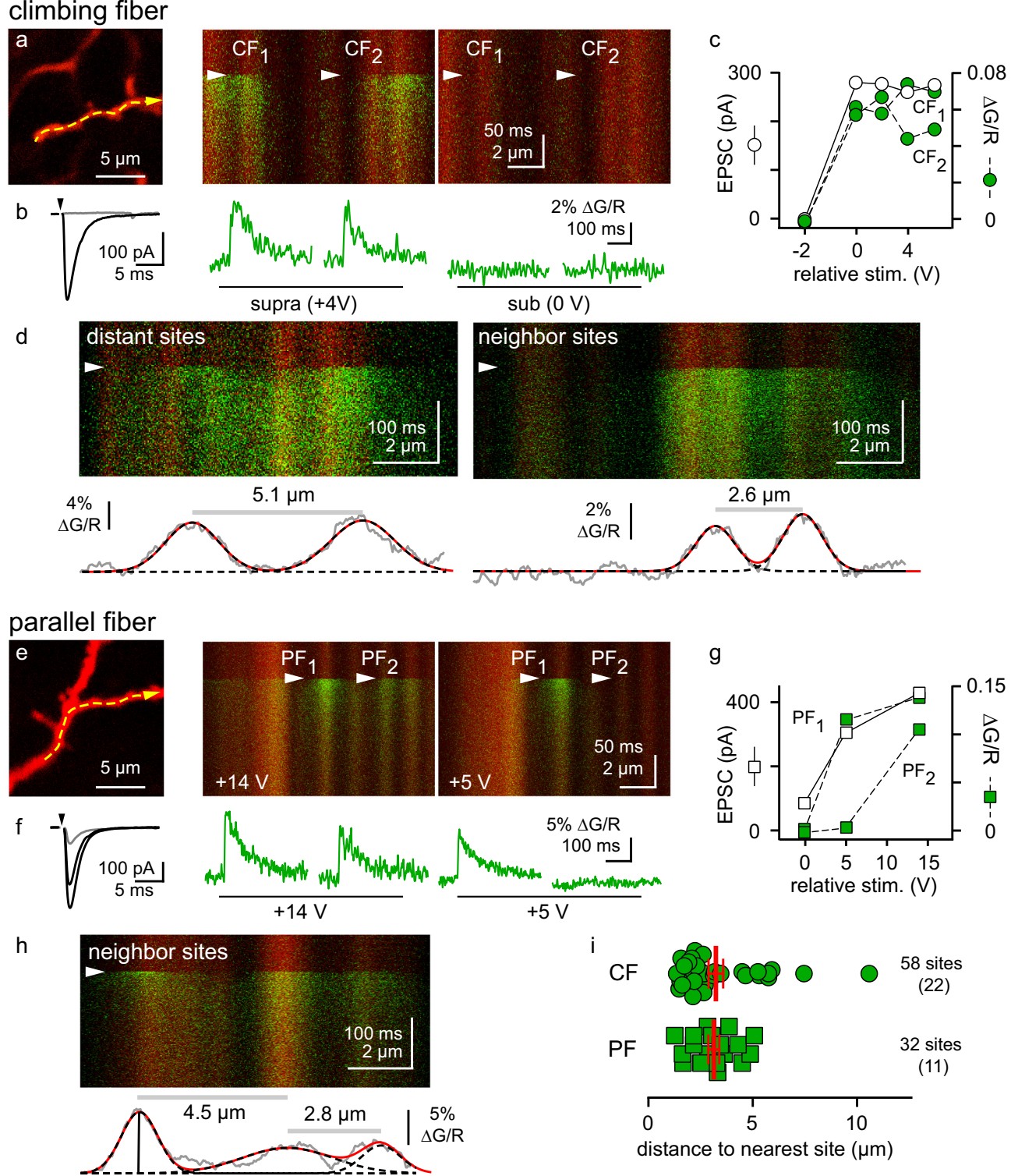

**Fig. 4 | Spillover activates AMPARs at multiple discrete sites. a** (left) MLI dendrite with two discrete spillover sites. **a** (right) Line scans were taken along the dashed line in response to supra- and subthreshold stimuli (arrowheads). **b** Corresponding CF EPSC and CaTs from (**a**). **c** EPSC (empty circles) or CaT (green circles) amplitude versus stimulus intensity from (**a**, **b**). **d** Example images and fluorescence profiles (gray traces) of distant (left; 5.1 μm) and neighbor (right; 2.6 μm) CF CaTs. CaTs spatial profile was determined using sums of Gaussian functions (black dashed lines; sum shown as red line) fit to the fluorescence profile of a single time point near the CaT peak (see text; 6 ms post-stimulus in this example). **e** Similar images to those shown in (**a**), but for PF responses.

**f** Corresponding PF EPSCs (black) are graded with stimulus intensity, while PF CaTs (green) are all-or-none at each site with varying thresholds. The gray EPSC represents the response when no more CaTs were visible on the scanned dendrite. **g** EPSC (empty squares) and CaT (green squares) amplitude versus stimulus intensity from (**e**, **f**). **h** Line scan with three discrete PF CaTs. **i** Summary of the distance between CaTs and their nearest neighbor site when evoked by CFs (3.2 ± 0.35 μm, $n = 58$ sites from 22 experiments) or PFs (3.1 ± 0.26 μm, $n = 32$ sites from 11 experiments). The average distance between sites was similar ($p = 0.4$, Two-tailed Mann–Whitney test). Data are shown as mean ± SEM. Source data are provided in the Source Data file: Source Data Figure4.xlsx.

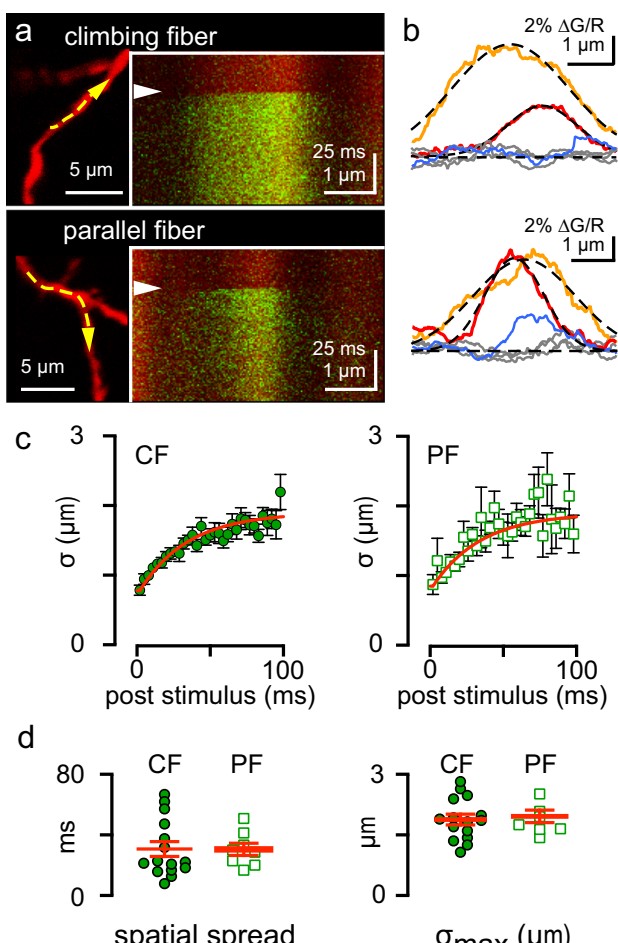

**Fig. 5 | Spillover and synaptic CaTs are confined to similar microdomains. a** Line scan path (dashed line) and line scan images of CaTs evoked by CF (top) or PF (bottom) stimuli (arrowhead). **b** Fluorescence profiles acquired in the 3 ms preceding CF or PF stimulation (gray), 1 ms (blue), 2 ms (red), and 25 ms (orange) post-stimulus. Profiles were fit with Gaussian functions (dashed lines) and the width (σ) was measured at each time point. **c** Plot of average σ values for CF ($n = 15$; left) and PF ($n = 8$; right) CaTs from 2 to 100 ms post-stimulus, fit with an exponential function (red line). **d** The time course of the CaT spatial spread (left, τ) and the maximum width (right, $\sigma_{max}$) do not differ between CFs and PFs (τ = CF: $31 \pm 4.9$ ms vs. PF: $30 \pm 4.0$ ms, $n = 15$ and 8, $p = 0.98$, and $\sigma_{max}$ = CF: $1.9 \pm 0.13$ μm vs. PF: $2.0 \pm 0.15$ μm, $n = 15$ and 8, $p = 0.72$, Two-tailed unpaired *t*-tests). Data are shown as mean ± SEM. Source data are provided in the Source Data file: Source Data Figure5.xlsx.

CF spillover (Fig. 7d; EPSC$_{subt}$/PF$_{alone}$: $0.54 \pm 0.043$ vs. $0.88 \pm 0.049$ vs. $0.92 \pm 0.075$; $n = 10, 6, 7$; $p < 0.001$, one-way ANOVA). To ensure valid comparisons, we adjusted the stimulus intensity of the PF pathway to recruit a few sites in all conditions (overlap: $97.0 \pm 10.9$ pA, non-overlap: $81.4 \pm 9.96$ pA; blind: $108 \pm 11$ pA, $p = 0.32$; one-way ANOVA; Supplementary Fig. 7a) and there was no difference in spillover EPSCs between groups (overlap: $331 \pm 20.0$ pA, non-overlap: $426 \pm 56.2$ pA, blind: $315 \pm 28.0$ pA; $p = 0.08$, One-way ANOVA; Supplementary Fig. 7c). Furthermore, there was no correlation between EPSC$_{subt}$/PF EPSC$_{alone}$ and the distance of recruited sites from the soma (overlap: $R^2 < 0.001$, $p = 0.99$; non-overlap: $R^2 = 0.19$, $p = 0.39$) nor CF EPSC$_{alone}$ amplitude (overlap: $R^2 = 0.001$, $p = 0.93$; non-overlap: $R^2 = 0.003$, $p = 0.92$; Supplementary Fig. 7c), suggesting that differential voltage control did not confound our interpretation. Together, these results show that CF spillover occludes PF synaptic transmission when both occur at the same site, and that separate afferent pathways can converge onto a common pool of AMPARs.

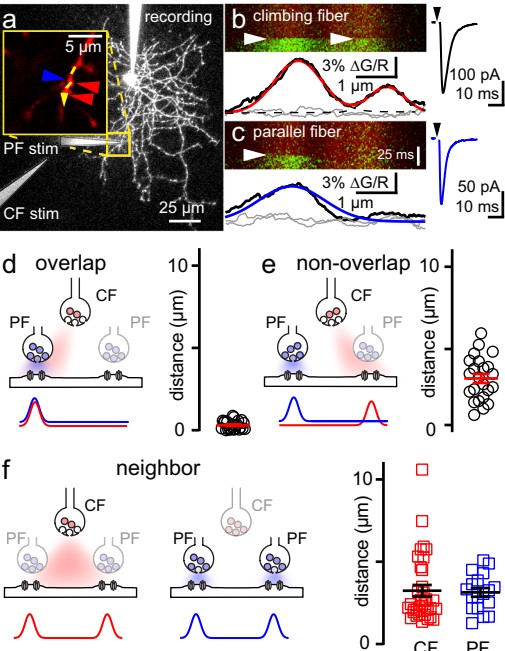

**Fig. 6 | Spatial overlap of CF and PF AMPAR CaTs. a** MLI image showing two-pathway stimulation setup for CF and PF stimulation. After localizing a CF CaT on an MLI dendrite (inset, red arrowheads), a second stimulating electrode was moved nearby to activate PF synapses on the same dendrite (inset, blue arrowheads). Line scans along the dashed yellow line revealed CaTs evoked from both pathways. **b, c** Average line scans (left) and EPSCs (right) from alternating CF or PF stimulation (white and black arrowheads, respectively). Fluorescence profiles near the peak in fluorescence (black, 3 ms post-stimulus in this example) are fit with Gaussian functions (CF, red; PF, blue). Fluorescence profiles preceding stimulation are gray. The left CF-evoked CaT overlaps with the PF-evoked CaT ($p > 0.05$, Two-sample KS test; see Supplementary Fig. 4). **d** Cartoon depiction of overlapping CF and PF AMPAR CaTs (as shown in **b, c**). The distance between peaks of the Gaussian fits to CF and PF overlapping sites ($0.31 \pm 0.05$ μm, $n = 24$ from 20 cells). **e** Cartoon depiction and distance between non-overlapping CF and PF AMPAR CaTs ($3.2 \pm 0.3$ μm, $n = 23$ from 20 cells). **f** Cartoon depiction and distance of neighboring CF (red; $3.2 \pm 0.35$ μm, $n = 34$) or PF (blue; $3.1 \pm 0.26$ μm, $n = 18$). Overlapping sites were more closely spaced than non-overlapping or neighboring sites ($p < 0.001$ for all comparisons). Non-overlapping sites from separate pathways (**e**) have similar spacing as neighbor sites evoked by CF or PF stimulation alone (**f**); $p > 0.99$ for all comparisons; Dunnett's T3 multiple comparisons test. Data are shown as mean ± SEM. Source data are provided in the Source Data file: Source Data Figure6.xlsx.

## GABA$_B$ receptors selectively inhibit synaptic inputs onto shared AMPARs

Excitatory synapses have varying sensitivity to neuromodulators, allowing independent regulation of distinct afferent inputs to a given neuron[57–60]. Here we show that CF and PF afferents can converge on a common population of synaptic receptors, raising the unexpected possibility that AMPARs at PF synapses could be active even when PF glutamate release is suppressed. In fact, repetitive PF activity suppresses PF glutamate release via presynaptic GABA$_B$ receptors, presumably due to GABA released from MLIs[61]. To test whether CF-mediated activation of CP-AMPARs at PF synapses is maintained when PF glutamate release is suppressed by GABA$_B$Rs[62], we applied the GABA$_B$ receptor agonist baclofen (3 μM) while alternating stimulation of overlapping CF- and PF-evoked CaTs (Fig. 8). Baclofen slightly reduced the CF spillover EPSC ($88 \pm 3.4\%$ of baseline; $p = 0.04$, Tukey's multiple comparisons test; Fig. 8a, c) with no effect on CF-evoked CaTs ($98 \pm 8.1\%$ of baseline; $p = 0.98$, Tukey's multiple comparison test; Fig. 8b, c) or the PPR ($0.25 \pm 0.050$ vs. $0.21 \pm 0.026$ after baclofen; $p = 0.58$, Tukey's multiple comparisons test). The reduction in the spillover EPSC was insensitive to the GABA$_B$R antagonist CGP55845

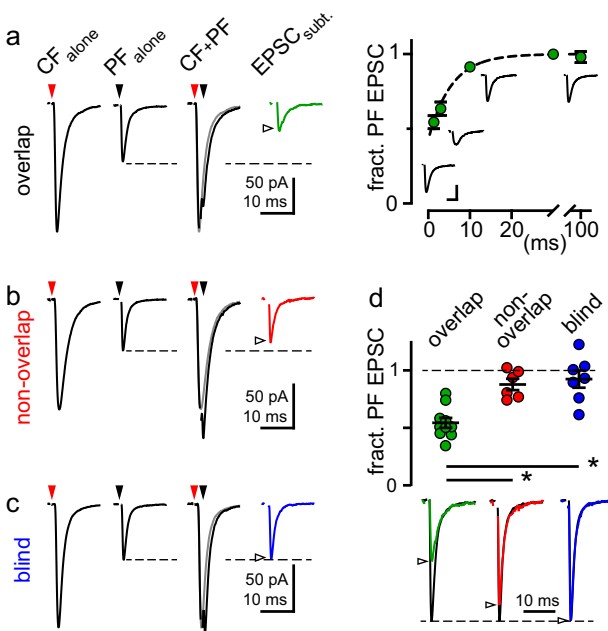

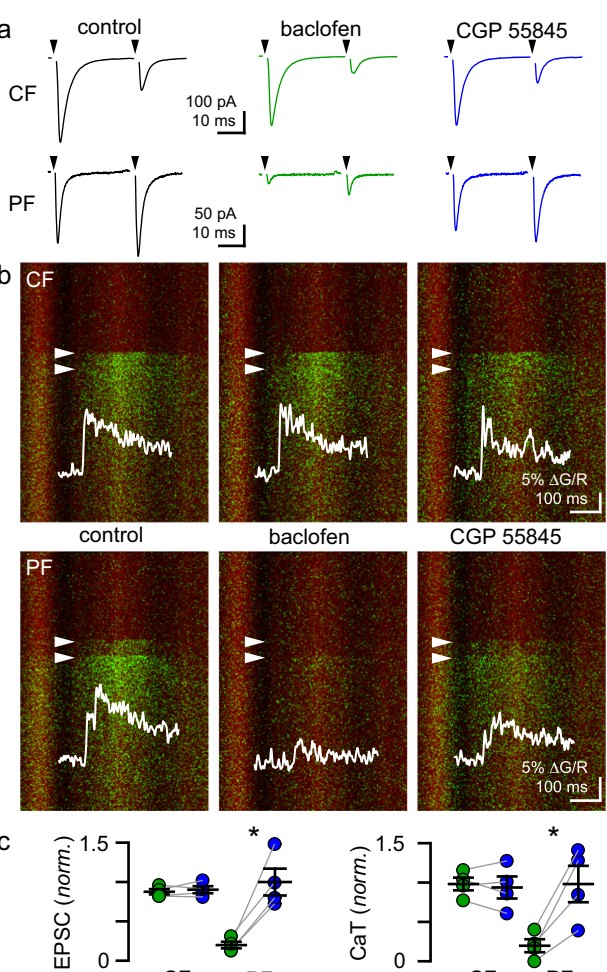

**Fig. 7 | PF EPSCs are occluded by CF EPSCs at overlapping sites. a** Control CF (red arrowhead) and PF (black arrowhead) EPSCs were evoked in isolation and then at a 1.3 ms ISI to generate a compound EPSC (CF + PF). The $CF_{alone}$ EPSC (gray) was subtracted from the compound EPSC (CF + PF) to isolate the subtracted PF component ($EPSC_{subt.}$; green trace). At overlapping sites, the $EPSC_{subt}$ was smaller than $PF_{alone}$ (empty arrowhead). $EPSC_{subt}$ recovered (right) with a time course of 5.5 ms (green dashed line, fit of $n = 4–10$). Scale bars: 25 pA, 5 ms. **b** Same analysis for CF and PF EPSCs evoked at non-overlapping sites (non-overlap $EPSC_{subt.}$; red) and when EPSCs were recruited randomly without fluorescence microscopy to guide electrode placement (**c**) (blind $EPSC_{subt.}$; blue). **d** The amplitude of $EPSC_{subt.}$ relative to $PF_{alone}$ EPSC for overlapping (green; $0.54 \pm 0.043$, $n = 10$), non-overlapping (red; $0.88 \pm 0.049$, $n = 6$), and blind (blue; $0.92 \pm 0.075$, $n = 7$) groups at 1.3 ms ISI. Below: Example $EPSC_{subt}$ from **a–c** (green, red, blue, empty arrowheads) normalized to $PF_{alone}$ EPSCs (black traces). Subtracted EPSCs from overlapping sites were smaller than at non-overlapping sites ($p = 0.03$) or blind sites ($p = 0.007$; Tukey's multiple comparisons test), whereas subtracted EPSCs from non-overlapping and blindly recruited sites were similar ($p = 0.85$; Tukey's multiple comparisons test). Asterisk denotes $p < 0.05$. Data are shown as mean $\pm$ SEM. Source data are provided in the Source Data file: Source Data Figure7.xlsx.

**Fig. 8 | GABA_BRs differentially inhibit CF and PF transmission onto shared AMPARs. a** Example CF and PF EPSCs before (control, black), after bath application of the GABA_BR agonist baclofen (3 μM; green), and the GABA_BR antagonist CGP55845 (1 μM; blue). **b** Line scans and CaTs corresponding to the EPSCs shown in (**a**). **c** (left) Normalized CF EPSCs in baclofen (green; $0.88 \pm 0.034$ of control, $p = 0.04$) and CGP55845 (blue; $0.90 \pm 0.045$ of control, $p = 0.8$). Suppression of PF EPSCs by baclofen (green, $0.20 \pm 0.042$ of control, $p = 0.003$) was reversed by CGP55845 (blue, $1.0 \pm 0.17$ of control, $p = 0.003$). **c** (right) CF and PF CaTs showed differential regulation by baclofen ($n = 4$). CF CaTs were unaffected by baclofen ($0.98 \pm 0.081$ of baseline, $p = 0.98$) or CGP55845 ($0.94 \pm 0.14$ of baseline, $p = 0.9$). PF CaTs were inhibited by baclofen ($0.20 \pm 0.083$ of control, $p = 0.008$) and inhibition was reversed by CGP55845 ($0.98 \pm 0.23$ of baseline, $p = 0.009$). Tukey's multiple comparisons test. Asterisk denotes $p < 0.05$. Data are shown as mean $\pm$ SEM. Source data are provided in the Source Data file: Source Data Figure8.xlsx.

(1 μM; $90 \pm 4.5\%$ of baseline; $p = 0.79$, Tukey's multiple comparisons test; Fig. 8a, c). In contrast, baclofen strongly reduced PF-evoked EPSCs ($20 \pm 4\%$ of baseline; $p = 0.002$, Tukey's multiple comparisons test) and CaTs ($20 \pm 4.2\%$ of baseline; $p = 0.008$, Tukey's multiple comparisons test; Fig. 8b, c) and increased the PPR of PF-evoked EPSCs ($1.5 \pm 0.23$ vs. $2.9 \pm 0.61$ after baclofen; $p = 0.03$, Tukey's multiple comparisons test), consistent with a presynaptic mechanism of action. Baclofen-induced changes in PF-evoked EPSC amplitude, CaT, and PPR were all reversed by CGP55845 (EPSC: $100 \pm 17.1$ of baseline, CaT: $98 \pm 23\%$ of baseline, PPR: $1.3 \pm 0.14$; $p = 0.99$, 0.99, 0.96, Tukey's multiple comparisons test; Fig. 8a, c). Together these results demonstrate that PF and CF glutamatergic transmission converging on a shared set of postsynaptic AMPARs are regulated independently, and that under conditions where canonical PF glutamate release is suppressed by neuromodulation, synaptic CP-AMPAR signaling can be hijacked by CF spillover.

## Discussion

Here we show that glutamate spillover originating from CFs activates spatially confined populations of CP-AMPARs along the dendrites of cerebellar MLIs. Although the temporal properties of CF-evoked spillover EPSCs differ from canonical PF EPSCs, the spatiotemporal properties of CF- and PF-CaTs shared striking similarities. Both CF- and PF-evoked CaTs are compartmentalized at microdomains at the same locations on MLIs, and consecutive activation at overlapping sites generates occlusion of PF EPSCs. These findings show that glutamate from CFs 'spills in' to PF-MLI synapses, illustrating an unexpected circuit motif where two independent afferent pathways share the same pool of receptors. Additionally, we show that presynaptic release onto shared postsynaptic AMPARs is independently regulated, illustrating that CP-AMPARs at PF synapses can be activated under conditions when PF synaptic release is suppressed.

The spillover of glutamate from CF-PC synapses to MLIs is well-described, but the location of AMPARs mediating spillover transmission has not been addressed. CF glutamate activates both AMPARs and NMDARs, with the latter comprised of NMDAR2B-containing receptors

located outside PF synapses[16,18,20,21,63]. Along with the lack of anatomically defined CF-MLI synapses[33,64–66], it was thus reasonable to predict that the AMPAR component would also be mediated by extrasynaptic AMPARs, and the highly compartmentalized CaTs generated by CF spillover was unexpected. While PF-MLI synapses are well-known to contain CP-AMPARs, the exact composition of both synaptic and extrasynaptic (assayed using somatic patches) AMPARs are regulated by activity[39,52,53]. Furthermore, the proportion of extrasynaptic and synaptic AMPARs has been debated. Early literature posited a large pool of mobile extrasynaptic AMPARs as a source for trafficking during synaptic plasticity[26], yet more recent data suggests AMPARs do not readily accumulate in extrasynaptic membranes[67]. Indeed, the density of extrasynaptic AMPARs on MLIs is very low[68]. While we cannot exclude that a small component of CF spillover EPSCs is mediated by a low density of extrasynaptic AMPARs, our results show that most of the spillover EPSC is generated by synaptic AMPARs. Our visualization of overlapping CaTs and the occlusion of PF-evoked EPSCs by CF spillover provides direct evidence that glutamate from CFs 'spills in' to activate AMPARs at PF synapses.

While the compartmentalization of CF CaTs reflects the postsynaptic properties of CP-AMPARs, the spillover [glutamate] transient differs from PF synaptic [glutamate] transient in its peak concentration and spatial spread[18,20,21]. The relatively low peak and prolonged time course of the spillover [glutamate] reflect the greater distance between CF release sites compared to PF sites and postsynaptic AMPARs. Early studies using conventional light microscopy reported transverse branches of CF axons passing close enough to MLIs that they were presumed synaptically connected[69,70]. Close apposition of transverse branches was subsequently confirmed, but the absence of synaptic release sites or markers of functional release makes these unlikely to be the source of CF spillover[65]. Rather, ascending CF axons that synapse with PCs are the most likely source of CF spillover. Interestingly, close appositions do exist between ascending CFs and MLIs, but they exhibit a specialized structure, distinct from electrical and chemical junctions, with a high density of Kv4.3 potassium channels on MLI membranes[64].

Synapse formation and maintenance is a complex, metabolically expensive process that requires the concerted interactions of pre- and postsynaptic molecules[71]. Transsynaptic interactions at conventional glutamatergic synapses result in precise alignment of release sites with areas of clustered AMPARs, facilitating efficient transmission[72]. Nonetheless, the crosstalk that we demonstrate here between separate afferent pathways produces a high-fidelity functional connection without the need to organize and maintain a canonical synapse. In fact, CF spillover is strong enough to drive MLI firing and mediate distinct patterns of Purkinje cell inhibition and disinhibition[20,21,37]. Robust crosstalk between CF and PF synaptic AMPARs is likely enabled by multivesicular release at CF-PC synapses that generates spillover to Bergmann glia, Golgi cells, and MLIs in a manner controlled by EAAT4 levels near PC synapses[34,73,74]. Recent work at hippocampal synapses shows that glutamate can activate high-affinity optical glutamate sensors in a radius >1.5 μm following quantal release, suggesting that crosstalk could be frequent across CNS synapses[31]. Future work will be needed to determine the distance between CF release sites and PF-MLI spillover sites, a distance that indicates the functional extent of glutamate crosstalk at low-affinity AMPARs after multivesicular release. Regardless of the distance, our results suggest that multivesicular release provides synaptic efficiency in the absence of nanodomain organization.

Distinct presynaptic afferents are typically synapse-specific and often segregated along the target cell's dendrites, as is the case for CF and PF synapses on PCs[46]. In contrast, CF-MLI transmission co-opts AMPARs located at PF synapses. To our knowledge, this is the first report of afferents arising from distinct sources converging onto a shared population of postsynaptic AMPARs. Detecting convergence of PF and CF afferents is not possible using electron microscopy, in contrast to other circuit motifs involving multi-contact synapses such as a single bouton innervating multiple dendrites or multiple PSDs on a single spine receiving input from multiple axons[33,75,76]. A combination of functional measures will likely be required to detect whether similar convergence between pathways occurs in other brain regions, especially in conditions that facilitate spillovers, such as intense synaptic activity, densely-spaced release sites, or multivesicular release.

Synaptic inputs from different sources that converge onto a common postsynaptic neuron can be differentially regulated by neuromodulators to enable dynamic gating of information pathways[57–60]. It is well established that activation of presynaptic GABA$_B$Rs potently inhibits the release of glutamate from PFs, whereas CFs are less sensitive to such presynaptic modulation[40,61,62,77]. Our results suggest the possibility that GABA release during intense MLI activity could dynamically gate the source of CP-AMPAR activation, switching the main source of CP-AMPAR activation at PF synapses to CFs. Such CF-mediated activation of MLI CP-AMPARs during strong PF activity could contribute to synaptic plasticity arising from coordinated stimulation of both pathways[36,78].

We can speculate on additional roles of this circuit motif in cerebellar signaling, beyond differential GABAergic modulation of release. In vivo studies have shown that CF activation drives the plasticity of PF-MLI synapses[36,78]; however, the cellular basis of such plasticity is unknown. Because Ca$^{2+}$-dependent signaling is spatially restricted along the lengths of MLI dendrites[39,79], the location of CF-MLI spillover sites may delimit specific PF-MLI synapses that are susceptible to CF-induced plasticity. Induction of such plasticity is likely to involve the activation of NMDARs by either or both pathways. How activation of NMDARs by either pathway contributes to synaptic plasticity will make an interesting subject of future studies.

Together these findings demonstrate a circuit motif wherein glutamate 'spill-in' from a morphologically unconnected afferent pathway co-opts synaptic receptors and allows activation of postsynaptic AMPARs even when canonical glutamate release is suppressed. This circuit motif shows that one-to-one connectivity of a given postsynaptic density with a presynaptic axon cannot be uniformly assumed and that functional measures are an important component of a full map of brain connectivity.

## Methods

All experimental preparations were performed using protocols approved by the Institutional Animal Care and Use Committee of the University of Alabama at Birmingham (IACUC-08767). Mice were housed at room temperature (25 °C) with a 12-h light/dark cycle and humidity between 40 and 60%. Mice were provided ad libitum access to food and water.

### Brain slice preparation

Parasagittal slices containing the cerebellar vermis were prepared from male and female wild-type C57BL/6 mice aged P27-P38 unless otherwise stated. Animals were anesthetized by isoflurane inhalation followed by intraperitoneal injection of 2, 2, 2-tribromoethanol (Avertin) and intracardial perfusion with an ice-cold cutting solution containing (in mM): 110 choline chloride, 2.5 KCl, 1.25 NaH$_2$PO$_4$, 0.5 CaCl$_2$, 7.0 MgCl$_2$, 25 NaHCO$_3$, 25 glucose, 11.5 sodium ascorbate, and 3 sodium pyruvate. Perfusion was followed by decapitation and dissection of the cerebellum. The cerebellum was glued to the cutting block of the vibratome stage (7000-SMZ, Campden Instruments) and kept submerged in an ice-cold cutting solution continuously bubbled with 95% O$_2$/5% CO$_2$ during slicing. Parasagittal slices (240 μM) containing the vermis were cut and incubated in (mM): 125 NaCl, 2.5 KCl, 1.0 NaH$_2$PO$_4$, 2.5 CaCl$_2$, 1.3 MgCl$_2$, 26.2 NaHCO$_3$, and 11 glucose, at 37 °C for 20 min before being stored at room temperature.

## Electrophysiology

Molecular layer interneurons (MLIs) were identified visually on an Olympus BX51WI microscope equipped with a 60×1.0NA objective (Olympus). Recorded cells were located in the anterior-most lobules of the vermis (lobules I/II and III) below the slice surface so that diffusion and connectivity more closely resembled that of intact tissue. Recording pipettes were pulled using a P-97 horizontal puller (Sutter) and filled with an internal solution containing (mM): 100 CsMeSO$_3$, 50 CsCl, 10 HEPES, 1 MgCl$_2$, 2 MgATP, 0.3 NaGTP, 5 QX 314, 0.03 Alexa 594, 0.2 Fluo-5F, and adjusted to pH 7.3 with CsOH. Filled pipettes had a tip resistance of 2.5–5.5 MΩ. Responses were measured using a Multiclamp 700B amplifier controlled by pClamp 10 software (Molecular Devices), filtered at 2–5 kHz, and digitized at 10–20 kHz (Digidata 1440). After obtaining a seal on an MLI membrane (at least 1 GΩ, typically 3–5 GΩ), the membrane was ruptured, and whole-cell recordings were made with a bath temperature of ~34 °C and a holding potential ($V_H$) of −60 mV. Series and input resistance ($R_s$ and $R_i$) were monitored during each sweep using a 10 mV step. Recordings were discarded if $R_s$ changed significantly (>20%) over the course of an experiment.

Climbing fibers (CFs) and parallel fibers (PFs) were stimulated using theta glass electrodes filled with external recording solution driven by Digitimer Constant Voltage (model DS2A Mk. II) or Current (model DS3A) Isolated Stimulators. Alexa 594 (5 μM) was added to the pipette solution to allow visualization while imaging. Stimulus strength varied between 0.5–50 V or 20–200 μA with a duration between 20–120 μs. Individual CFs could be stimulated by placing a stimulating electrode near the base of the soma or primary dendrite of a PC near the recorded MLI. PFs were activated by placing a stimulating electrode above the dendrites of the recorded MLI. Sweeps were collected at 0.05–0.1 Hz for the CF pathway and 0.1–0.2 Hz for the PF pathway. In dual-pathway stimulation experiments, each pathway was stimulated at 0.1 Hz.

## Two-photon Ca²⁺ imaging

The internal recording solution for simultaneous 2 P imaging and electrophysiological recording included the Ca$^{2+}$ indicator Fluo 5 F (200 μM) to detect sites of Ca$^{2+}$ influx. Two-photon excitation was achieved using a Chameleon Vision or Ultra II pulsed Ti:Sapphire lasers (Coherent) tuned to 810 nm for simultaneous imaging of cell morphology and Ca$^{2+}$ influx. Laser power was modulated via a Pockels cell (Model 350-80 Electro-Optic Modulator, ConOptics). Images were acquired on an Olympus BX51WI microscope equipped with a 60×1.0NA objective (Olympus). The point spread function (PSF) of the system at 810 nm was measured using green 100 nm diameter beads (Tetraspeck™ Microspheres, Invitrogen). The full width at half maximum of the PSF (Supplementary Fig. 3) was 445 ± 11 nm laterally and 1893 ± 102 nm axially ($n = 6$). Pairs of photomultiplier tubes (PMTs) collected light from epi- and transfluorescence pathways. Both pathways contained a 565 nm long pass beam splitter (565lxpr; Chroma), a GaAsP PMT (H7422P-40; Hamamatsu) with a 525/50 bandpass filter (ET525/50 m; Chroma), and a multi-alkali PMT (R3896; Hamamatsu) with a 595/50 bandpass filter (ET595/50 m; Chroma). Prairie View software (Bruker Corporation, V5.7) was used for the acquisition of imaging data.

## Localizing and imaging Ca²⁺ transients

Imaging was initiated after at least 10 min of whole-cell dialysis to allow Alexa 594 and Fluo-5F to equilibrate. After isolating a CF or PF EPSC, sequential frame scanning occurred at 2–5 Hz by visually inspecting for increased Fluo-5F fluorescence coincident with evoked EPSCs. PF-evoked Ca$^{2+}$ transients (CaTs) were easily identified below the stimulating electrode, while CF-evoked CaTs were difficult to localize due to their wide range of intensities and sparseness throughout the MLI dendritic arbor (Fig. 1). CF CaTs were localized in approximately half of the recordings where CF EPSCs were successfully isolated, but their absence does not imply they do not exist because we could not scan the entire dendritic tree. A potential limitation of using Ca$^{2+}$ imaging to localize synaptic sites is that the spatial extent of Ca$^{2+}$ signals can overestimate the spread of receptor activation as a result of the Ca$^{2+}$ indicator diffusing from the initial receptor "point source", since all Ca$^{2+}$ indicators act as mobile buffers that can shuttle Ca$^{2+}$ along the dendrite. We minimized this possibility by using low-affinity dyes and our estimates of spatial compartmentalization are consistent with previous reports[38,39].

After observing a putative CF- or PF-evoked CaT, the dendrite where the CaT was observed was magnified and a path for line scans (rate of 0.5 or 1 kHz) was drawn along the dendrite. Line scans were triggered using TTL outputs driven by pClamp software. Scans were triggered on 2–4 consecutive electrophysiology sweeps, with at least 1 min between bouts.

## Axotomy

Methodology and analysis of axotomy was based on previously published work[51]. We used male and female wild-type C57BL/6 mice aged P17-21 with a smaller dendritic arbor and pronounced axon. The intracellular solution in these experiments contained (in mM): 130 K-gluconate, 5 KCl, 0.5 EGTA, 10 HEPES, 4 MgATP, 0.4 NaGTP, and 0.015 Alexa 594. Axons were visually identified, and a cutting location was selected <30 μm from the soma in an area away from dendrites. Axotomy was reliably achieved with 5–10 line scans across the cutting location at 200 mW. The fast and slow capacitance time constants ($\tau_{fast}$ and $\tau_{slow}$) were determined before and after axotomy by fitting the decay of the current response to a −10 mV voltage step with a biexponential function. $C_m$ fast and $C_m$ slow were then calculated as follows:

$$C_m x = \frac{A_x \tau_x}{-10\ mV} \tag{1}$$

where $A_x$ and $\tau_x$ are the amplitude and time constant of the fast or slow capacitance component. Successful axotomies were determined by a (1) marked reduction in $C_m$, (2) fading (within 5 min) of the fluorescence within the axon compartments proximal (but downstream) of the cut, and (3) formation of a bleb at the cut site.

## Analysis of Ca²⁺ transients

Line scan sweeps from each recording were aligned using the peaks of the fluorescence profile of the Alexa 594 (red) channel to correct for drift. The aligned images were then analyzed individually and averaged. CaTs are shown as ΔG/R and calculated using:

$$\frac{\triangle G}{R} = \frac{(G - G_0)}{R} \tag{2}$$

where $G_0$ is equal to the average Fluo-5F fluorescence preceding stimulation, G is the magnitude of Fluo-5F fluorescence at a given time point, and R is Alexa 594 fluorescence. ΔG/R was calculated along the length of every line scan in a sweep. ΔG/R as a function of time was calculated from the average ΔG/R of the ten pixels surrounding the spatial peak of the CaT. CaTs were then smoothed using a Gaussian filter (width = 10 ms, α = 4.2).

ΔG/R as a function of space was analyzed by filtering individual scans (Gaussian, width = 1 μm, α = 0.5) from multiple averaged sweeps and fitting them with one or more Gaussian functions:

$$\frac{\Delta G}{R}(x) = A e^{-(x-b)^2/2\sigma^2} \tag{3}$$

where A is the amplitude of the transient at that time point, b is the position of the peak along the scan in μm, and σ is equal to 34% of the transient peak in μm. When multiple CaTs were present, the scan was fit using the sum of a number of Gaussians equal to the number of sites present.

To determine if CF and PF CaTs recruited along the same stretch of dendrite were overlapping, fluorescence profiles were cropped 2–5 σ on either side of the peak ΔG/R(x) and the cumulative value for ΔG/R was calculated for both transients. Cumulative ΔG/R was then normalized and compared using a two-sample Kolmogorov–Smirnov test. Sites were considered overlapping when $p > 0.05$. Of 22 recordings where there were putatively overlapping CF and PF sites, 2 were excluded from analysis due to failure of the KS test ($p < 0.05$).

### Pharmacology

All recordings were made in the presence of picrotoxin (PTX; 100 μM, Abcam) to block GABA$_A$ receptors, except for dual pathway experiments where gabazine (GBZ; 10 μM, Abcam) was used instead to reduce the external concentration of DMSO. NMDA receptor responses were blocked with R-CPP (5 μM, Abcam) in all recordings. In experiments involving dual-pathway stimulation, blockers of L-type Ca$^{2+}$ channels (nifedipine; 10 μM, Tocris), T-type Ca$^{2+}$ channels (TTA-P2; 5 μM, Alomone Labs), and CB1 receptors (AM251; 1 μM, Cayman Chemical Company) were applied along with R-CPP. AMPARs were blocked using either NBQX (10 μm, Abcam) or NASPM trihydrochloride (50–100 μM, Tocris). GABA$_B$ receptors were activated using the agonist (R)-baclofen (3 μM, Tocris) and antagonized using CGP55845 hydrochloride (1 μM, Abcam). mGluR1 receptors were blocked with CPCCOEt (100 μM, Abcam). All drugs were applied via the external recording solution, except for QX-314 chloride (5 mM, Abcam), which was used in all experiments to block voltage-dependent Na$^+$ channels and was added to the internal solution.

### Analysis

Electrophysiological and time series Ca$^{2+}$ imaging data were analyzed using Axograph X software. Initial image inspection and processing was performed using ImageJ. This included averaging line scan data from PMTs in the epi- and transfluorescence pathways for each sweep, and background subtraction for Z-series images. The distance of a CaT from the soma was estimated by drawing a straight line from the site of the CaT to the soma on a maximum Z-series projection of the image in ImageJ. Likewise, the distance of an MLI soma from the PCL was also estimated by drawing a straight line from the soma to the PCL. While Purkinje cells were not labeled in these experiments, their position could be determined by: (1) placement of the stimulating electrode, (2) faint PC autofluorescence visible when increasing contrast and decreasing the brightness threshold in ImageJ, and (3) the position of axonal projections in the case of basket cells.

Custom MATLAB scripts were used to align images, calculate time series data from line scans, fit images with Gaussian functions, and plot the position of CaTs and MLI soma within the ML.

### Statistics and reproducibility

Statistical analysis and plots were performed using GraphPad Prism 9. All data were shown as mean ± SEM unless otherwise indicated. Datasets comprised of two groups were analyzed using either a student's t-test, Welch's t-test, or the Mann–Whitney test as appropriate. No statistical method was used to predetermine the sample size. Datasets with more than one group were analyzed with a one-way ANOVA with or without repeated measures. Tukey's or Dunnett's multiple comparisons test was used if the means of all groups were compared to one another or if all groups were compared to the mean of a single group, respectively. Welch's ANOVA test and Dunnett's T3 multiple comparison test were used if the standard deviation between groups was unequal.

### Reporting summary

Further information on research design is available in the Nature Portfolio Reporting Summary linked to this article.

## Data availability

The data generated in this study are provided as a Source data file with this paper. Raw data will be provided upon request.

## Code availability

The custom-made MATLAB scripts used for analysis can be downloaded from https://github.com/rpennock/Spill-in-Manuscript-MATLAB

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

## Acknowledgements
We thank Mary Seelig for technical assistance throughout this study, members of the Wadiche labs, Dr. Scott Cruikshank, and Dr. Anastasios Tzingounis for helpful discussions on the manuscript. This research was supported by funding from the National Institutes of Health (F32NS110154 to R.L.P., R01NS113948 to J.I.W., and R01NS105438 to L.O.-W.) and the Civitan International Research Center (Civitan Emerging Scholars Award, R.L.P.).

## Author contributions
R.L.P., L.O.-W., and J.I.W. designed the study, and R.L.P., L.T.C., and X.Y. performed electrophysiology and imaging experiments. R.L.P. and J.I.W. analyzed data. R.L.P., L.O.-W., and J.I.W. wrote and edited the paper.

## Competing interests
The authors declare no competing interests.
