## [Peer Review File · Nature Communications]

REVIEWER COMMENTS

Reviewer #1 (Remarks to the Author):

The MS by Pennock et al investigates a fundamental and difficult issue in synaptic neuroscience, which is the characterisation of the activation of transmitter receptors outside of point synapses, by spillover mechanisms. The MS focuses on glutamatergic transmission in the cerebellum from the climbing fibre to molecular layer interneurons, which has previously been reported to be mediated exclusively by spillover. The authors combine electrophysiological and imaging techniques, the latter exploiting the fact that calcium-permeable AMPA receptors are recruited. The central and important conclusion is that the receptors activated via spillover by the climbing fibre are largely synaptic receptors included within parallel fibre synapses.

It is my opinion that the design, experiments, analysis and presentation are all of very high quality. Some of the experiments are remarkable - for instance the control for a depolarisation confound comparing occlusion of CaTs at overlapping and non-overlapping PF and CF sites (Fig. 7).

I very strongly support publication. I'm curious whether the authors have examined or will examine activation of NMDA receptors. There is some evidence that their activation might govern the plasticity of parallel fibre synapses on interneurons, and there is also the unresolved issue of where these NMDARs are expressed and how they are activated by parallel fibres.

I have struggled to find anything requiring correction or improvement in the MS.

One aspect that I think could be quantified and discussed in more detail is the relative strength of the two converging signals. Fig. 3C suggests that CaT fluorescent signals are equivalent for CF and PF activation. Is this representative? It's worth pointing out that although the recent Matthews et al paper suggests long-range spillover effects at standard synapses, that conclusion is probably subject to a couple of unresolved artefacts (poor spatial resolution of the imaging and, especially, of the uncaging, whose duration allows extensive diffusive exchange and spread of cage and products, potentially increasing the amount of glutamate uncaged.)

I'm not sure the authors report the total number of CaTs and the number of CaTs per MLI recorded (the spatial plot of Fig. 1H suggests one). What was the yield for finding CF CaTs?

I did observe a minor issue in Fig. 3D. The legend has a couple of typos "NAPSM" and "CF-", while the label should presumably be "fractional response" not "fractional block".

Reviewer #2 (Remarks to the Author):

This study analyzed the spatial location and temporal dynamics of AMPAR-mediated calcium responses triggered by glutamate spillover from climbing fibers and its influence on the PF-molecular layer interneuron synapses in the cerebellum. Investigators found that glutamate spillover from climbing fibers activates Ca²⁺-permeant AMPARs at PF-MLI synapses and characterized the location and mapping of this activation with high spatial and temporal precision. They visualized the locations where Ca²⁺-permeant AMPARs were activated by glutamate spillover from climbing fibers in molecular layer interneurons and investigated the extent of spatial overlap between CF and PF AMPAR-mediated calcium transients. In addition, the study also showed that GABAergic activity specifically regulates PF synaptic transmission but does not affect glutamate spillover from climbing fibers. Overall, the experimental design and execution of this study were well-planned and carried out with advanced techniques and analysis methods. The results indicate that glutamate spillover from climbing fibers to molecular layer interneurons shares the same Ca²⁺-permeant AMPARs as PF inputs and influences their synaptic activity, even when PF synaptic release is suppressed. It suggests crosstalk between distinct afferent fibers on the same target. As a result, this study is well-suited for publication in Nature Communications. However, before publishing, the authors may want to consider the following points for further improvement:

1. It would be informative to address how often (frequently) overlapping CF and PF AMPAR calcium transients (CaTs) were observed in comparison to other responses, such as non-overlapping or neighboring CaTs. Additionally, the proportion of overlapping CF and PF AMPAR CaTs among all recordings would provide further insights into the prevalence of this phenomenon. For example, what percentage of MLIs displayed overlapping CF/PF CaTs, 10%, 50%, or 100%? For those cells displaying overlapping CaTs, did this typically sparse, (occurring at just one or two locations) or was this widespread throughout the dendritic arbor?
2. In Figure 3C, the CaTs (ΔG/R) triggered by CF stimulation and those triggered by PF stimulation were similar in amplitude. Does this suggest the glutamate concentration in the synaptic cleft from each source (CF and PF) is similar or saturating? It would be interesting to understand how glutamate spillover (mediated by CF), with its relatively low concentration and prolonged time course, can result in similar effects to those of PF synaptic activity. It would be valuable to estimate the glutamate

concentration in the local area at PF-MLI synapses, either through PF stimulation or CF stimulation, or both.

3. It would be interesting to know what, if any, role glutamate transporters play in this process. This group has previously shown that spillover from CF to MLIs depends on the expression of EAAT4, which shows patterned expression in the cerebellum. Does the prevalence of overlapping CaTs correlate with EAAT4/aldolase-C expression? Does inhibiting glutamate transport increase the number or amplitude of overlapping CF/PF CaTs?

4. Lastly, it would be beneficial to explore the physiological significance of this new type of circuit, where glutamate spill-in activates synaptic receptors that are already occupied by other inputs, in the cerebellum circuitry. Some discussion of how this type of synaptic signaling is relevant to the cerebellar circuit would be helpful.

In conclusion, the study provides valuable insights into the spatial location and temporal resolution of AMPAR-mediated Ca responses activated by glutamate spillover and its impact on PF-MLI synapses in the cerebellum. The authors are encouraged to consider the points prior to publication.

Reviewer #3 (Remarks to the Author):

In the present manuscript, the Wadiche lab shows that at cerebellar climbing fiber (CF) to molecular layer interneuron (MLI) synapses spill-over transmission recruits AMPA receptors that are actually located at parallel fiber (PF) synapses. This study is based on two-photon imaging in cerebellar slices and patch-clamp electrophysiology.

The study is very well executed and the data are analyzed and presented with care. The finding is of interest and will advance not only the field of cerebellar physiology, but will be of wider interest to an audience interested in synaptic transmission and the various forms that it can assume. There is one major problem that needs to be addressed to exclude a participation of metabotropic glutamate receptors (mGluRs).

The authors correctly point out that what they describe here is an interesting circuit motif, in which synaptic responses are evoked at an input site that is not contacted by the glutamate releasing terminals. (This differs from previously described spill-over transmission, where the activated receptors are not located at classic synapses with both pre- and postsynaptic structural components.) If I remember correctly, though, a similar scenario has been described by Knopfel at CF-Purkinje cell synapses, where glutamate released from CF terminals can bind to mGluRs anywhere along the dendritic tree and cause (calcium) responses, as long as there is a co-occurring dendritic calcium

transient, but can be pharmacologically identified as an mGluR response (Yuan et al., Mol. Cell. Neurosci. 35, 2007; see also prior studies by Nusser et al., Neuroscience 61, 1994 and Petralia et al., Neuropharmacology 37, 1998). Similar to the findings in the new Wadiche study, this response is blocked by AMPA receptor inhibition. This points to a scenario where CFs can release glutamate, which spills over to PF synapses where CP-AMPA receptor activation creates a calcium transient that then allows for mGluR signaling upon agonist binding to the receptors. Whether such scenario is likely in the present study is unclear given the relatively brief dendritic responses, but the authors should exclude this possibility by applying an mGluR antagonist. Also, the signaling scenario described in the Knopfel work should be discussed to provide the reader with a complete overview of related literature.

Minor: on p. 8, it should read 30um, not uM.

Response to Reviewers

Reviewer #1:

The MS by Pennock et al investigates a fundamental and difficult issue in synaptic neuroscience, which is the characterisation of the activation of transmitter receptors outside of point synapses, by spillover mechanisms. The MS focuses on glutamatergic transmission in the cerebellum from the climbing fibre to molecular layer interneurons, which has previously been reported to be mediated exclusively by spillover. The authors combine electrophysiological and imaging techniques, the latter exploiting the fact that calcium-permeable AMPA receptors are recruited. The central and important conclusion is that the receptors activated via spillover by the climbing fibre are largely synaptic receptors included within parallel fibre synapses.

It is my opinion that the design, experiments, analysis and presentation are all of very high quality. Some of the experiments are remarkable - for instance the control for a depolarisation confound comparing occlusion of CaTs at overlapping and non-overlapping PF and CF sites (Fig. 7).

I very strongly support publication.

We appreciate the reviewer's positive assessment of our work. Those experiments were particularly difficult, and we are glad to see them appreciated.

1) *I'm curious whether the authors have examined or will examine activation of NMDA receptors. There is some evidence that their activation might govern the plasticity of parallel fibre synapses on interneurons, and there is also the unresolved issue of where these NMDARs are expressed and how they are activated by parallel fibres.*

This is a very interesting question that we now mention in the discussion. We are currently initiating a thorough investigation of the localization of NMDARs activated by CFs and PFs, and how that activation relates to plasticity. This will require a complete set of experiments in a subsequent manuscript.

2) *One aspect that I think could be quantified and discussed in more detail is the relative strength of the two converging signals. Fig. 3C suggests that CaT fluorescent signals are equivalent for CF and PF activation. Is this representative?*

Good point. In fact, when comparing the amplitude of CF and PF CaTs occurring at overlapping input sites the CF CaTs are slightly larger, on average ($p=0.04$, $n=24$ overlapping sites, paired t test; see graph below). This is discussed in more detail in our response to reviewer #2, and now included in the text and Supplementary Fig 5. When we compared the ratio of CF and PF CaTs to the charge of their respective EPSCs, we found that the CaT/EPSC ratio is greater for PFs that generate a higher [glutamate] compared to CF spillover, presumably from activation of more CP-AMPA. Please also see response to Rev2, pt.2,

3) I'm not sure the authors report the total number of CaTs and the number of CaTs per MLI recorded (the spatial plot of Fig. 1H suggests one). What was the yield for finding CF CaTs?

We did not report the total number of CaTs or the fraction of MLIs where a CaT was found. We located CF-evoked CaTs in approximately half of attempted recordings. However, it is unclear whether this fraction is due to the lack of CaTs or the inability to find a CF CaT. We identify CaTs by observing the patched MLI in live scan mode and while monitoring for increased fluorescence coincident with the evoked EPSC. A limitation of 2P imaging is that we only see the cross section of the MLI dendritic arbor within a narrow focal plane (PSF < 2 μm , Supplementary Fig 3). The MLI dendritic arbor does not project very far into the transverse plane, however there are times where evoked CaTs were localized to segments of dendrite that were optically isolated from the rest of the dendritic arbor (for examples see Supplementary Fig 1A, Supplementary Fig 6B). This isolation could result from undulations in the morphology of the dendrite that place small segments in a different focal plane from the rest of the dendrite, a distal location on a dendrite, or a location on a branch that is partially projecting in the transverse plane. Evoked CaTs also occur with a variety of fluorescence intensities. Together, this means that some CaTs are quite dim and may be located on sections of dendrite that are difficult to image because of their orientation and position within the dendritic arbor. Thus, we cannot interpret a failure to localize a CaT as an indication it does not exist. This is now clarified in the 'Localizing and Imaging Ca²⁺ Transients' section in the Methods.

4) I did observe a minor issue in Fig. 3D. The legend has a couple of typos "NAPSM" and "CF-", while the label should presumably be "fractional response" not "fractional block".

These mistakes have been corrected in the figure and figure legend.

Reviewer #2:

This study analyzed the spatial location and temporal dynamics of AMPAR-mediated calcium responses triggered by glutamate spillover from climbing fibers and its influence on the PF-molecular layer interneuron synapses in the cerebellum. Investigators found that glutamate spillover from climbing fibers activates Ca²⁺-permeant AMPARs at PF-MLI synapses and characterized the location and mapping of this activation with high spatial and temporal precision. They visualized the locations where Ca²⁺-permeant AMPARs were activated by glutamate spillover from climbing fibers in molecular layer interneurons and investigated the extent of spatial overlap between CF and PF AMPAR-mediated calcium transients. In addition, the study also showed that GABAergic activity specifically regulates PF synaptic transmission but does not affect glutamate spillover from climbing fibers. Overall, the experimental design and execution of this study were well-planned and carried out with advanced techniques and analysis methods. The results indicate that glutamate spillover from climbing fibers to molecular layer interneurons shares the same Ca²⁺-permeant AMPARs as PF inputs and influences their synaptic activity, even when PF synaptic release is suppressed. It suggests crosstalk between distinct afferent fibers on the same target. As a result, this study is well-suited for publication in Nature Communications.

We thank the reviewer for the positive assessment of our work.

1) It would be informative to address how often (frequently) overlapping CF and PF AMPAR calcium transients (CaTs) were observed in comparison to other responses, such as non-overlapping or neighboring CaTs. Additionally, the proportion of overlapping CF and PF AMPAR CaTs among all recordings would provide further insights into the prevalence of this phenomenon. For example, what percentage of MLIs displayed

overlapping CF/PF CaTs, 10%, 50%, or 100%? For those cells displaying overlapping CaTs, did this typically sparse, (occurring at just one or two locations) or was this widespread throughout the dendritic arbor?

A limitation of the technique is that we cannot fully map the location of all spillover sites from a given CF onto an MLI (please see also Rev 1 point 3), or all of the overlapping CF/PF sites, so it is not possible for us to quantify the frequency of overlapping and non-overlapping sites. We have clarified this in the Methods. In total, in 20 out of 69 MLI recordings with a CF CaT we also recruited an overlapping PF CaT, but in those 49 recordings without overlapping PF CaTs, we did not try to recruit overlapping sites.

In terms of the prevalence of overlapping CaTs, we reiterate that when multiple CF-evoked CaTs are recruited along the length of a dendrite, the spacing of those CaTs is consistent with that of PF-MLI synaptic sites (Fig 4I) and the confinement of CF-evoked CaTs is identical to that of PF sites (Fig 5). Together with the presence of overlapping sites and occlusion of EPSCs at overlapping sites, we propose that CF-evoked CaTs are *exclusively* mediated at PF-MLI synapses. In other words, we speculate that every CF-evoked AMPAR-mediated CaT most likely originates within a PF-MLI synapse.

In some experiments where an overlapping CF/PF-evoked CaT was present there was also another CF-evoked CaT in the same image that did not overlap with a PF site (e.g. Fig 6A-C and Supplementary Fig 6A). CF CaTs were sparse rather than widespread throughout the dendritic arbor. Further, these experiments were designed to determine whether overlapping sites *could* occur, not whether we could evoke an overlapping PF site at *all* CF sites. The presence of CF spillover sites without overlapping PF sites is likely explained by the PF innervating that site not being recruited by our stimulating pipette, perhaps cut during slicing. We are cautious about interpreting negative data.

2) In Figure 3C, the CaTs ($\Delta G/R$) triggered by CF stimulation and those triggered by PF stimulation were similar in amplitude. Does this suggest the glutamate concentration in the synaptic cleft from each source (CF and PF) is similar or saturating? It would be interesting to understand how glutamate spillover (mediated by CF), with its relatively low concentration and prolonged time course, can result in similar effects to those of PF synaptic activity. It would be valuable to estimate the glutamate concentration in the local area at PF-MLI synapses, either through PF stimulation or CF stimulation, or both.

Good point. We have now addressed this comment by testing the sensitivity of EPSCs to the low affinity AMPAR antagonist kynurenic acid (500 μ M) while stimulating release from CFs or PFs. Inhibition of CF spillover by KYN was indeed greater than that of PF synaptic release (as previously reported in young rats ¹, demonstrating that the peak [glutamate] from CFs is lower than from PFs. These data are now included in the text and as part of Supplementary Fig 5

When comparing the amplitude of CF and PF CaTs occurring at overlapping sites, the CF CaTs are slightly larger, on average ($p=0.04$, $n=24$ overlapping sites, paired t test; see graph below). To understand this slight difference, we also took into account the different time course of glutamate at the same CP-AMPARs. PFs generate a high but brief [glutamate] transient, whereas CF spillover generates a lower but prolonged [glutamate] transient, leading to EPSCs with different kinetics ^{1,2}. Thus, we measured the ratio of CaTs to the integral of the respective EPSCs at overlapping sites in experiments with near-minimal PF stimulation. In this new analysis, we find the CaT/EPSC ratio is greater for PFs compared to CF spillover, presumably due to activation of more CP-AMPARs with a higher [glutamate].

We have included this data in new Supplementary Fig 5. However, there are caveats to this interpretation. While we can minimize PF stimulation to single or few sites, we do not know how many sites contribute to CF EPSCs, nor whether CI-AMPA receptors also contribute to EPSCs. In cases where multiple CaTs evoked by a single pathway were present (e.g. the CF and PF pathway in Supplementary Fig 6) those CaTs were summed for this analysis. Further, AMPARs activated by low concentrations may have lower conductance³. Thus, we remain cautious about overinterpreting the amplitude of CaTs using synaptic versus spillover stimulation.

3) It would be interesting to know what, if any, role glutamate transporters play in this process. This group has previously shown that spillover from CF to MLIs depends on the expression of EAAT4, which shows patterned expression in the cerebellum. Does the prevalence of overlapping CaTs correlate with EAAT4/aldolase-C expression? Does inhibiting glutamate transport increase the number or amplitude of overlapping CF/PF CaTs?

We agree it is interesting to understand how glutamate transporters contribute to this process. As noted in the methods, we performed these experiments in lobules I/II & III of the vermis due to the low expression of EAAT4 to optimize the chances of detecting CF-MLI spillover⁴. We intend to follow up with a comparison in high EAAT4 regions to address physiologically relevant differences in glu transporters, also expanding analysis to Golgi cells that also receive CF spillover⁵. However, this analysis is complicated by the need to visualize high and low EAAT4 areas with Venus during live Ca²⁺ imaging.

Blocking transporters increases the amplitude and duration of CF EPSCs. Accordingly, our preliminary data indeed shows that TBOA increases the amplitude of CF CaTs and recruits CaTs at neighboring sites, consistent with glutamate diffusing further from its point source. We intend to include this data in future work in conjunction with EAAT4 expression.

4) Lastly, it would be beneficial to explore the physiological significance of this new type of circuit, where glutamate spill-in activates synaptic receptors that are already occupied by other inputs, in the cerebellum circuitry. Some discussion of how this type of synaptic signaling is relevant to the cerebellar circuit would be helpful.

Thanks for this point, we have added more discussion on the implications of this circuit motif to cerebellar physiology in the discussion section titled 'Afferent convergence onto shared synaptic AMPARs'.

Reviewer #3

In the present manuscript, the Wadiche lab shows that at cerebellar climbing fiber (CF) to molecular layer interneuron (MLI) synapses spill-over transmission recruits AMPA receptors that are actually located at parallel fiber (PF) synapses. This study is based on two-photon imaging in cerebellar slices and patch-clamp electrophysiology.

The study is very well executed and the data are analyzed and presented with care. The finding is of interest

and will advance not only the field of cerebellar physiology, but will be of wider interest to an audience interested in synaptic transmission and the various forms that it can assume.

We appreciate the positive assessment of our work.

1) There is one major problem that needs to be addressed to exclude a participation of metabotropic glutamate receptors (mGluRs)... Whether such scenario is likely in the present study is unclear given the relatively brief dendritic responses, but the authors should exclude this possibility by applying an mGluR antagonist.

To address this possibility, we performed additional experiments recording CF-evoked CaTs before and after application of the mGluR1 antagonist CPCCOEt (100 μ M). This blocks postsynaptic mGluRs in MLIs⁶. There was no significant effect of CPCCOEt on CF CaT amplitude. These findings have been added to the manuscript in the 'Mapping CF-evoked Ca²⁺ transients on MLI dendrites' section of the results and are included in Supplementary Fig 1G.

2) If I remember correctly, though, a similar scenario has been described by Knopfel at CF-Purkinje cell synapses, where glutamate released from CF terminals can bind to mGluRs anywhere along the dendritic tree and cause (calcium) responses, as long as there is a co-occurring dendritic calcium transient, but can be pharmacologically identified as an mGluR response (Yuan et al., *Mol. Cell. Neurosci.* 35, 2007; see also prior studies by Nusser et al., *Neuroscience* 61, 1994 and Petralia et al., *Neuropharmacology* 37, 1998). Similar to the findings in the new Wadiche study, this response is blocked by AMPA receptor inhibition... Also, the signaling scenario described in the Knopfel work should be discussed to provide the reader with a complete overview of related literature.

We now cite relevant papers from the Knopfel and Otis labs in the results sections to explain the reasoning behind these experiments. This section of text is included in the response to point #1.

3) Minor: on p. 8, it should read 30um, not uM.

This has been corrected in the manuscript.

References:

1. Szapiro, G. G. & Barbour, B. B. Multiple climbing fibers signal to molecular layer interneurons exclusively via glutamate spillover. *Nat. Neurosci.* **10**, 735–742 (2007).
2. Coddington, L. T., Rudolph, S., Vande Lune, P., Overstreet-Wadiche, L. & Wadiche, J. I. Spillover-mediated feedforward inhibition functionally segregates interneuron activity. *Neuron* **78**, 1050–1062 (2013).
3. Rosenmund, C., Stern-Bach, Y. & Stevens, C. F. The Tetrameric Structure of a Glutamate Receptor Channel. *Science* **280**, 1596–1599 (1998).

4. Malhotra, S. *et al.* Climbing Fiber-Mediated Spillover Transmission to Interneurons Is Regulated by EAAT4. *The Journal of Neuroscience* vol. 41 8126–8133 Preprint at <https://doi.org/10.1523/jneurosci.0616-21.2021> (2021).
5. Nietz, A. K., Vaden, J. H., Coddington, L. T., Overstreet-Wadiche, L. & Wadiche, J. I. Non-synaptic signaling from cerebellar climbing fibers modulates Golgi cell activity. *Elife* **6**, e29215 (2017).
6. Karakossian, M. & Otis, T. Excitation of cerebellar interneurons by group I metabotropic glutamate receptors. *J. Neurophysiol.* **92**, 1558–1565 (2004).

REVIEWERS' COMMENTS

Reviewer #1 (Remarks to the Author):

I am entirely satisfied by the authors' responses.

Reviewer #2 (Remarks to the Author):

The authors have effectively addressed all the comments and suggestions raised during the previous review. Their thorough responses, as well as the incorporation of additional supplemental data, analyses, and textual revisions, have substantially enhanced the manuscript. I particularly appreciate the authors for their diligent revisions, particularly the sensitivity test of EPSCs to the low-affinity AMPAR antagonist kynurenic acid (500 μ M) while stimulating release from CFs or PFs. This new supplemental data successfully addresses the comments regarding the similarity of CaTs (DG/R) triggered by CF stimulation and those triggered by PF stimulation.

The revised manuscript has significantly improved in terms of clarity, scientific rigor, and overall quality. The authors have successfully addressed all previous concerns, making the manuscript highly suitable for publication in Nature Communications.

Reviewer #3 (Remarks to the Author):

All my previous concerns have been appropriately addressed.